# Secretory Phospholipase A2 and Interleukin-6 Levels as Predictive Markers of the Severity and Outcome of Patients with COVID-19 Infections

**DOI:** 10.3390/ijms24065540

**Published:** 2023-03-14

**Authors:** Stanislav Urazov, Alexandr Chernov, Oleg Popov, Natalya Klenkova, Natalya Sushentseva, Irina Polkovnikova, Svetlana Apalko, Kseniya Kislyuk, Dragana Pavlovich, Andrey Ivanov, Sergey Shcherbak

**Affiliations:** 1City Hospital 40 of Saint Petersburg, 197706 St. Petersburg, Russia; 2Bioenergetics Department of Life Sciences, Ben-Gurion University, Beer Sheva 84105, Israel; 3Federal State Budgetary Institution of Science “Institute of Experimental Medicine”, 197376 St. Petersburg, Russia; 4Center for Clinical and Laboratory Diagnostics, Federal State Budgetary Military Educational Institution of Higher Education “Military Medical Academy named after S.M. Kirov”, 194044 St. Petersburg, Russia; 5Department of Postgraduate Medical Education, Federal State Budgetary Educational Institution of Higher Professional Education “Saint-Petersburg State University”, 199034 St. Petersburg, Russia

**Keywords:** COVID-19 infection, SARS-CoV-2, phospholipase A2, interleukin-6, laboratory blood markers, prognosis, severity, death, patients, correlation, odd ratio

## Abstract

Coronavirus disease (COVID-19) has become a global pandemic. COVID-19 patients need immediate diagnosis and rehabilitation, which makes it urgent to identify new protein markers for a prognosis of the severity and outcome of the disease. The aim of this study was to analyze the levels of interleukin-6 (IL-6) and secretory phospholipase (sPLA2) in the blood of patients regarding the severity and outcome of COVID-19 infection. The study included clinical and biochemical data obtained from 158 patients with COVID-19 treated at St. Petersburg City Hospital No. 40. A detailed clinical blood test was performed on all patients, as well as an assessment of IL-6, sPLA2, aspartate aminotransferase (AST), total protein, albumin, lactate dehydrogenase (LDH), APTT, fibrinogen, procalcitonin, D-dimer, C-reactive protein (CRB), ferritin, and glomerular filtration rate (GFR) levels. It was found that the levels of PLA2, IL-6, APTV, AST, CRP, LDH, IL-6, D-dimer, and ferritin, as well as the number of neutrophils, significantly increased in patients with mild to severe COVID-19 infections. The levels of IL-6 were positively correlated with APTT; the levels of AST, LDH, CRP, D-dimer, and ferritin; and the number of neutrophils. The increase in the level of sPLA2 was positively correlated with the levels of CRP, LDH, D-dimer, and ferritin, the number of neutrophils, and APTT, and negatively correlated with the levels of GFR and lymphocytes. High levels of IL-6 and PLA2 significantly increase the risk of a severe course by 13.7 and 2.24 times, and increase the risk of death from COVID-19 infection by 14.82 and 5.32 times, respectively. We have shown that the blood levels of sPLA2 and IL-6 increase in cases which eventually result in death and when patients are transferred to the ICU (as the severity of COVID-19 infection increases), showing that IL-6 and sPLA2 can be considered as early predictors of aggravation of COVID-19 infections.

## 1. Introduction

Coronaviruses (CoV) are the causative agents of acute severe respiratory syndrome (SARS-CoV), which first caused a global epidemic in 2002 [1]. In December 2019, a new form of coronavirus (SARS-CoV-2) was recorded in China, which caused a global pandemic of coronavirus disease (COVID-19), which spread to 228 countries in 2020–2023 [2]. To date, the pandemic has caused more than 669 million infections and more than 6.7 million deaths worldwide [3].

Currently, there are more than 1000 different genetic lines of SARS-CoV-2. Most SARS-CoV-2 mutations do not manifest phenotypically. Epidemiological significance is characteristic only for individual lines. To analyze the prevalence, clinical significance, and biological properties (pathogenicity, contagiousness, and neutralizing activity of antibodies) of various variants of the virus, the WHO proposed to identify variants of concern (VOC) and variants of interest (VOI). VOC lines are characterized by biological properties that increase the contagiousness and pathogenicity or reduce the neutralizing activity of antibodies [4]. To date, there are several VOC lines: the α-line (PANGO B.1.1.7, clade 501.YV,1 69–70del, Y144del), first registered in the UK in September 2020 [5]; the β-line (PANGO B.1.351), first discovered in South Africa in May 2020, containing nucleotide substitutions N501Y, K417N, and E484K in the S-protein [6]; the γ-line (PANGO p.1, 484K.V2), containing the key mutation E484K in the S-protein, first isolated from samples of patients from the state of Rio de Janeiro (Brazil) in November 2020 [7]; the δ-line (PANGO B.1.617.2), containing mutations L452R and T478K in the S-protein, first discovered in India in October 2020 [8]; and the Omicron line (PANGO B.1.1.529), containing S477N mutations and first detected in South Africa and Botswana in November 2021 [9].

The clinical spectrum of COVID-19 manifestations varies from the asymptomatic form of the disease and the appearance of symptoms of acute respiratory viral disease: fever (90%), cough (80% of cases), shortness of breath (30% of cases), fatigue (40% of cases), chest congestion (20%), sore throat, runny nose, decreased sense of smell and taste, conjunctivitis, pneumonia, life-threatening complications (including acute respiratory distress syndrome (ARDS)), sepsis, septic shock, and multiple organ failure [4]. These pathological conditions most often lead to mortality in patients of working age (59.7 ± 13.3 years) with chronic diseases such as arterial hypertension (23.7–30%), diabetes mellitus (16.2%), metabolic syndrome, coronary heart disease (5.8%), chronic obstructive pulmonary disease (COPD), nicotine addiction, inflammatory intestinal diseases, and oncological pathology [10,11,12]. The clinical picture in patients at risk is characterized by the development of a “mutual burden syndrome”, accompanied by progressive respiratory and heart failure, which ultimately aggravates their condition and leads to labor losses, early disability, and high mortality. The mortality rate of hospitalized patients ranges from 15% to 20%, and is higher in those in need of intensive care [13].

Central to the pathophysiology of COVID-19 disease is immune dysfunction with a pronounced uncontrolled generalized systemic inflammatory reaction due to increased production of inflammatory cytokines, known as a cytokine storm (CS). A CS is accompanied by fever, cytopenia, hyperferritinemia, abnormal liver parameters, coagulopathy, and lung damage (including ARDS) [14]. In all these conditions, the cytokines IL-1β, IL-18, IFN-γ, and IL-6 are the main mediators of hyperinflammation. COVID-19-associated CS is a unique form of hyperinflammatory reaction requiring the development of criteria for its establishment [15]. In addition, lymphocytopenia and neutrophilia are common, with a significant decrease in the number of CD8^+^ T cells, CD4^+^ T cells, and natural killer cells (NK) [16]. Biomarkers of lipid metabolism occupy a key place among clinically significant analytes [17]. The most common markers include glucose; triglycerides; cholesterol; saturated and omega-polyunsaturated fatty acids; and high, low, and very low density lipoproteins and their enzymes [18]. Such lipid enzymes include sPLA2 [19]. sPLA2 is a family of lipolytic enzymes that perform diverse functions and are involved in the pathogenesis of a wide range of diseases, e.g., inflammatory arthritis [20], bronchial asthma [21], ARDS [22], atherosclerosis [23], cancer [24], obesity, [25], sepsis [26], and bacterial [27] and viral infections [28], as well as COVID-19 [29]. At the same time, the prognostic value of PLA2 as a biomarker of the severity and outcome of COVID-19 infection in patients remains debatable.

In this regard, COVID-19 infected patients with chronic diseases are particularly in urgent need of immediate diagnosis and rehabilitation. It is extremely important to combat this pandemic to study the pathogenesis of this disease and identify new protein targets that may turn out to be highly sensitive and specific prognostic markers of the severity and outcome of the disease, which will allow personalizing medical rehabilitation and therapy programs. The definition of such new biomarkers should be accessible for clinical diagnosis, as cheaply and informatively as possible. The availability of biomarkers is determined by the possibility of their rapid detection in tissue samples or biological fluids of the patient using methods used in laboratory diagnostics.

The aim of this work was to analyze the levels of interleukin-6 and secretory phospholipase in the blood of patients depending on the severity and outcome of the COVID-19 infection.

## 2. Results

Initially, we studied the severity and outcome in 158 patients with COVID-19 infections.

Depending on the severity of the COVID-19 infection, patients were divided into groups with mild 38.6% (*n* = 61), average 23.4% (*n* = 37), and severe 38.0% (*n* = 60) severity. The survival rate (discharge) of patients was 80.4% (127/158) and the mortality was 19.6% (31/158). Survival among men was 81.6%, while among women it was 78.3% (Table 1).

Statistically significant differences depending on age among discharged and deceased patients and among men and women were not found. The results are presented as the arithmetic means ± standard deviations.

Then, the influence of the cardiological history and some clinical indicators, such as transfer to the ICU, BMI, number of days from the onset of the disease from biobanking, and maximum CT on the severity of COVID-19 infection, was determined (Table 2).

The results presented in Table 2 show that the transfer to the ICU, the number of days from the onset of the disease, and the CT index statistically significantly increased with an increased severity of the disease. The severity of COVID-19 infection was affected by pulmonary circulatory disorders, the incidence of patients with hypertension, coagulopathy, and iron deficiency anemia.

A comparative analysis of IL-6 and Noas-2 levels was performed in patients with laboratory parameters such as APTT, AST, GFR, LDH, CRP, IL-6, sPLA2, D-dimer, procalcitonin, ferritin, hematocrit, lymphocytes, neutrophils, and eosinophils, depending on the severity of COVID-19 infection (Figure 1).

The results shown in Figure 1 show that the levels of APTV, AST, CRP, LTK, IL-6, D-dimer, and ferritin, as well as the number of neutrophils, significantly increased in patients with mild and severe COVID-19 infections (Figure 1). On the contrary, hematocrit, GFR, lymphocyte, and eosinophil levels were statistically significantly reduced in these groups of patients compared to the values in patients with a mild infection. The PLA2 level also significantly increased in mild to moderate COVID-19 infections, but it did not change in patients with severe infections. These data on changes in the level of PLA2 in the blood show that this enzyme can be considered as an early marker of exacerbation of COVID-19 infection.

IL-6 and PLA2 levels were also analyzed depending on the severity of the disease according to computed tomography (Figure 2).

The results shown in Figure 2 show that the levels of IL-6 and PLA2 increase statistically significantly with an increase in the CT score. Moreover, the increase in PLA2 is almost 2 times higher than the increase in IL-6 at the beginning of the disease, when the CT score changes from 0 to 1. On the contrary, the level of IL-6 increases two-fold at the height of the disease, when the CT score increases from 2 to 3. In addition, the dependence of the PLA2 level on the severity of infection is also confirmed by the presence of a sufficiently high correlation coefficient between the two (r = 0.311171, *p* = 0.00069).

An analysis of laboratory tests was carried out depending on the levels of IL-6 and sPLA-2.

Patient samples were also divided into groups with high and low levels of IL-6 and PLA2. The criteria for dividing IL-6 and PLA2 into low- and high-level groups were the values of their first quartile (Q1); values below Q1 were assigned to the low-level group and all values above Q1 were assigned to the high-level group of analytes (Table 3 and Figure 3).

The results shown in Table 3 show that the increased in the levels of APTT, AST, LDH, CRP, D-dimer, ferritin, and the number of neutrophils are statistically significantly different in groups with low and high concentrations of IL-6. On the contrary, hematocrit, GFR, lymphocyte, and eosinophil levels significantly decrease in the group of patients with medium and high concentrations of IL-6. The results shown in Table 3 show that the increase in the level of CRP and the decrease in lymphocytes are statistically significantly different in groups with low, medium, and high concentrations of PLA2. A statistically significant increase in LDH and D-dimer levels was observed between groups of patients with low and medium, and low and high concentrations of PLA2. An increase in APTT, ferritin, and the number of neutrophils and a decrease in GFR were statistically significantly observed between the groups of patients with low and high concentrations of PLA2. A statistically significant increase in AST levels was observed only between groups of patients with low and medium concentrations of PLA2. The number of eosinophils and hematocrit decreased between the groups of patients with low and high and medium and high severity of COVID-19 infection.

Correlation coefficients were calculated between the levels of IL-6, sPLA2 concentrations, demographics, clinical parameters, and history of concomitant diseases (Table 4).

The results in Table 4 show that the levels of IL-6 and PLA2 are statistically significantly correlated with the transfer of patients to the ICU, CT score, the presence in the anamnesis of uncomplicated diabetes mellitus, and iron deficiency anemia in patients. In addition, IL-6 showed a weak but significant correlation with weight. PLA2 also showed a weak but significant correlation with obesity, and the level of IL-6 correlates with the presence of pulmonary circulatory disorders, pulmonary artery pressure, peripheral vascular disorders, coagulopathy, renal insufficiency, rheumatoid arthritis, collagen, and vascular diseases.

Correlations between the levels of PLA2 and IL-6 and the studied laboratory parameters were also analyzed (Table 5 and Figure 3).

The results in Table 6 and Figure 3 show that the level of IL-6 is statistically significantly positively correlated with the levels of PLA2, ASIA, AST, DV, SKI, and the laboratory score, and is negatively correlated with the number of lymphocytes and the PAC index. The level of PLA2 also correlates positively with the level of IL-6, SKI, and the laboratory score, and negatively correlates with the number of lymphocytes and the PAC index. Moreover, the correlation values for IL-6 were higher than for sPLA2.

At the final stage, the endpoint (outcome of COVID-19 infection) was analyzed depending on the severity, the levels of IL-6, PLA2, ARTT, AST, LDH, CRB, GFR, procalcitonin, D-dimer, ferritin, and hematocrit, the number of lymphocytes, leukocytes, neutrophils, and eosinophils, and the laboratory score (Table 6 and Table 7).

The results presented in Table 7 show that the probability of death from a COVID-19 infection is greatly increased in patients with a severe course (*p* < 0.0001); high levels of IL-6, PLA2, ARTT, AST, LDH, CRB, d-dimer, ferritin, and neutrophil counts; and low levels of GFR, lymphocytes, and hematocrit. The dependence of the outcome of COVID-19 infections on the level of PLA2 is also confirmed by the presence of a sufficiently high correlation coefficient between them (r = 0.310402, *p* = 0.0000721).

Finally, we calculated the odd ratio (OR) for the severity and the outcome of COVID-19 infections using our analytes (Figure 4).

The results presented in Figure 4 show that increased levels of LDH, IL-6, and procalcitonin are the factors most predictive of the development of a severe or fatal COVID-19 infection in patients. Elevated levels of PLA2, ACT, D-dimer, APTT, and hemocrit also predicted a severe or fatal COVID-19 infection in patients, with odd ratios of 2.24, 5.46, 5.68, 4.71, 6.39 and 5.32, 4.14, 2.85, 4.51, 12.81, respectively. A combination of elevated levels of PLA2 and IL-6 increased the risk of a severe infection and death of the patients from COVID-19 by 20.0 and 9.68 times.

## 3. Discussion

Secretion of PLA2-IIA aggravates damage to tissues and organs of the whole organism [17,22,30]. This may contribute to the severity and number of deaths due to COVID-19 infections [31]. Indeed, the results of [32,33] showed that in severe COVID-19 infections, plasma phospholipid levels decrease and the levels of lysophospholipids (lyso-PL), acylcarnitines, and non-esterified unsaturated fatty acids increase. These changes in the lipid profile indicate an increase in the activity of sPLA2 lysing membrane phospholipids in COVID-19 infections [31]. At the same time, increased levels of phosphatidylcholine 16:1_22:6 (AUC = 0.97), phosphatidylethanolamine 18:1_20:4 (AUC = 0.94), AK (AUC = 0.99), and oleic acid (AUC = 0.98) in 103 patients with COVID-19 infections correlated with the severity of the disease. There is a suppression of the biosynthesis of tyrosine, phenylalanine, tryptophan, and aminoacyl-tRNA [8].

In this study, it was found that sPLA2 levels increase in seriously ill patients, especially in the eventual case of death, and positively correlate with the severity and outcome of COVID-19 infections (Table 2, Table 5, Table 6 and Table 7, Figure 1, Figure 2, Figure 3 and Figure 4). These results are consistent with the data of a single study, which also showed that high levels of sPLA2-IIA in blood plasma correlate with its activity (r^2^ = 0.84, *p* = 1.2 × 10^−13^) and the severity of COVID-19 infection in 127 patients. In the group of patients who died from COVID-19, sPLA2-IIA levels could reach 1020 ng/mL, and were higher (89.3 ng/mL) than in those with severe (17.9 ng/mL) and mild (9.3 ng/mL) courses of the disease and those without coronavirus infection (8.9 ng/mL). Additionally, using a regression analysis model, sPLA2-IIA and urea nitrogen (BUN), at levels of 10 ng/mL and 16 mg/dL, respectively, were determined as the main clinical parameters for predicting mortality from COVID-19 infections with high accuracy (AUROC 0.93–1.0) and a sensitivity of 75.4% [31]. In another study, sPLA2 levels were elevated (269 ± 137.3 ng/mL, *p* = 0.01) in the blood plasma of 14 children with severe COVID-19 infections compared with those with asymptomatic (2.0 ± 3 ng/mL) and mild (23.0 ng/mL) cases. At the same time, the level of sPLA2 was increased (*p* = 0.04) in patients in the acute phase of the disease (540 ± 510 ng/mL) compared with the recovery period (2 ± 1 ng/mL). No correlations were found between sPLA2 and CRB and D-dimer levels and the leukocyte count [34]. At the same time, positive correlations between sPLA2-IIA levels and NEWS2 indicators and glucose levels, and negative correlations between urea creatinine, glomerular filtration rate, hematocrit, and hemoglobin saturation were determined, which also confirms the dependence of sPLA2-IIA on the severity of COVID-19 infection [14,35].

In addition, the severity of COVID-19 infection is positively correlated with the level of viremia and the number of apoptotic cells expressing PS. Consequently, such cells will be destroyed by sPLA2, which further increases systemic inflammation [36]. Inhibition of sPLA2-induced cell damage can be considered as a new approach against uncontrolled inflammation and cytokine storms.

This study also shows that the level of IL-6 increases in seriously ill patients and in the cases of unfavorable outcomes and it is correlated with the severity and death in patients (Table 2, Table 5, Table 6 and Table 7, Figure 1, Figure 2 and Figure 3). Our results are consistent with the research of other authors evaluating the prognostic significance of IL-6 in COVID-19 infections [37]. For example, in a retrospective single-center study conducted on 728 patients with COVID-19, the prognostic significance of elevated IL-6 levels for assessing mortality and a severe disease course was studied [38]. Using the logistic regression analysis of Cox in this study, the adjusted ratio of mortality risks and the severity of the disease in patients was evaluated. The authors concluded that elevated IL-6 levels may serve as an independent risk factor for severity and mortality in patients with COVID-19 [38]. A meta-analysis deserves attention, in which prognostic factors and the significance of elevated IL-6 levels in this pathology were determined in 1426 patients infected with COVID-19 [39]. It was shown that the best predictor of mortality or severe COVID-19 infection in patients is IL-6 (OR = 11.6460, 95%CI = 2.8123–48.2277), which predicts the onset of endpoints (outcomes) with an accuracy of 80.8%. Consequently, the level of IL-6 can serve as a prognostic marker of severe course, and especially where the outcome could be death, in patients with COVID-19 infections [39]. In an earlier study, we developed a model for assessing the risk of cytokine storms (CS) in 458 patients with COVID-19 infections [40]. The patients were divided into two groups, comparable in age. The first group consisted of 100 (21.8%) patients with clinical and radiological features characterizing a stable course of the disease of moderate severity and the second group consisted of 358 (78.2%) people with progressive moderate, severe, and extremely severe cases of the disease. When conducting a comparative analysis of clinical, instrumental, and laboratory data from the selected groups of patients, we found significant differences in the dynamics of the index on the NEWS scale, the absolute number of lymphocytes, and the levels of CRP, ferritin, D-dimer, and IL-6 between the groups, which can serve as the most important indicators characterizing the development of a CS. Using the method of constructing classification trees, the threshold levels for risk factors for the development of a CS were identified. We performed a comprehensive assessment of the risk of CSH by ranking the indicators, which, in accordance with the rank of prognostic significance obtained by the method of constructing “classification trees”, at the beginning of CSH therapy was as follows: dynamics of the index on the NEWS scale; blood IL-6 level above 23 pg/mL; blood CRP level equal to or above 50 mg/L; absolute number of lymphocytes less than 0.72 × 10^9^/L; positive test result for coronavirus RNA (SARS-CoV-2); and an age of 40 years or older. These biomarkers can be used as criteria for assessing the risk of a CS. An increase in the frequency of CS cases correlates with an increase in the number of risk factors (correlation coefficient Rg = 0.91, *p* < 0.001). The following risk categories are identified for the practical application of our predictive model: category 1 (0–1 factor): the risk of CSH is practically absent; category 2 (2–3 factors): the risk of CSH increases sharply to 55%, a 35.5-fold increase compared to category 1; category 3 (4 or more factors): the risk of CSH reaches 96%, an 718-fold increase 718 compared to category 1. The results obtained are consistent with the assessment of risk factors for CS in COVID-19 by other authors [41,42] and allowed us to justify the choice of therapeutic tactics with early prescriptions of proactive anti-inflammatory therapy and anticoid plasma convalescents for patients with a high risk of CS.

## 4. Materials and Methods

### 4.1. Patients

The retrospective cohort study included clinical and biochemical data obtained from 158 patients (98 men and 60 women aged 51.2 ± 11.6 years), who showed a positive test results for the presence of SARS-CoV-2 RNA by nucleic acid amplification in polymerase chain reaction (PCR), treated at the budget healthcare institution “City Hospital No. 40 of St. Petersburg Resort administrative district”, the boarding house “Zarya”, and “City Hospital No. 40” from 1 September 2020 to 15 October 2021. The average follow-up time of the clinical course was 10 days. The inclusion criteria were (1) age over 18 years and (2) a positive result of a PCR test for SARS-CoV-2 RNA. The exclusion criteria were (1) age under 18 years, (2) severe course of COVID-19 infection, (3) impaired consciousness, (4) unstable hemodynamics, (5) severe course of other somatic diseases, (6) severe course of oncological diseases, and (7) acute phase of other inflammatory and immune diseases.

The main endpoint of the study was biological death. Additional endpoints were desaturation and transfer to the ICU. The study was approved by the Ethics Expert Council of St. Petersburg City Hospital No. 40 and No. 205, dated 2 November 2021, and was conducted in accordance with the general principles of observational research.

We determined the severity of COVID-19 infection in our patients based on assessment of their clinical status using the NEWS-2 scale, the degree of lung involvement on CT, the blood level of serum and plasma biomarkers, and their laboratory score.

The design of the study is presented in Figure 5.

### 4.2. Clinical Methods and Treatment of Patients with COVID-19

All patients were admitted for inpatient treatment in the emergency room to the infectious diseases department. It was mandatory to provide the entire volume of medical services in accordance with medical and economic standards and in accordance with the version of clinical recommendations in force at the time [4]. Data were collected from the patients, e.g., their epidemiological history, and the presence of clinical symptoms (cough, shortness of breath, fever, fever, weakness, loss of sense of smell and taste, and heaviness in the chest). We also conducted an objective examination of patients with an assessment of hemodynamic parameters, an assessment of the respiratory system (HR, HR, BP, and SpO_2_), and an assessment of the NEWS scale recommended for use for patients with COVID-19 [43]. On the day of admission or the next day, biomaterial was collected for laboratory tests and an electrocardiogram (ECG) was performed. Computed tomography (CT) of the chest organs was performed with an assessment of the form of the disease on a 4-digit scale without intravenous contrast enhancement (CT-1, CT-2, CT-3, and CT-4). The bilateral lower lobe; the peripheral, perivascular, multilobular character; numerous peripheral seals in the form of “frosted glass” with a rounded shape of various lengths; flattening of the interlobular interstitium in the type of a “cobblestone pavement”; areas of consolidation; symptoms of an air bronchogram, etc. [44], were mainly assessed and, if necessary, additional instrumental methods were used.

According to the national recommendations for the diagnosis and treatment of COVID-19 [4], we used the following classification of COVID-19 according to severity:Mild course: body temperature < 38 °C, cough, weakness, and a sore throat. Absence of criteria for moderate and severe courses.Moderate course: body temperature > 38 °C, respiratory rate > 22/min, shortness of breath during physical exertion, changes in CT (radiography) typical of a viral lesion, SpO_2_ < 95%, and serum CRP > 10 mg/L.Severe course: respiratory rate > 30/min; SpO_2_ ≤ 93%; PaO_2_/FiO_2_ ≤ 300 mmHg; decreased level of consciousness; agitation; unstable hemodynamics (systolic blood pressure less than 90 mmHg or diastolic blood pressure less than 60 mmHg, diuresis less than 20 mL/h); changes in the lungs in CT (radiography) typical of a viral lesion; arterial blood lactate > 2 mmol/L; and qSOFA > 2 points.Extremely severe course: persistent febrile fever; ARDS; acute respiratory failure (ARF) with the need for respiratory support (invasive ventilation); septic shock; multiple organ failure; changes in the lungs on CT (X-ray) typical of a critical viral lesion or ARDS.

Treatment of COVID-19 infections and its complications included antiviral drugs, prevention of hypercoagulation and DIC syndrome, symptomatic treatment, and oxygen therapy. In patients with a progressive course of the disease, for the prevention or treatment of a cytokine storm (CS), standard therapy was supplemented with the appointment of pathogen-induced plasma convalescents, anti-cytokine drugs, interleukin-6 receptor inhibitors (IL-6) (tocilizumab, olokizumab, and levilimab), IL-1 (kanakinumab and RH104), JAK kinases (tofacitinib, ruxolitinib, and baricitinib), tyrosine kinase Bcr–Abl (radotinib), and, in some cases, glucocorticosteroids [40]. According to the indications, staged respiratory therapy, antibacterial therapy, treatment of sepsis and septic shock, extracorporeal detoxification and hemocorrection, and extracorporeal membrane oxygenation were performed.

A statement of biological death was made by the ICU doctor. The transfer of the patient in accordance with the indications specified in the clinical recommendations of the ICU was carried out and registered at the conclusion of the examination by the on-duty resuscitator. A decrease in blood saturation during dynamic observation was recorded by the medical staff and the attending physician: below 95% in the air and the moment of supply of moistened oxygen through a mask or nasal cannulas in a volume of 5 L per minute, or until saturation of more than 95% with a constant flow of oxygen. Desaturation was reported to the duty officer and attending physicians and noted in the observation sheet. The doctor gave the command to start oxygen insufflation.

### 4.3. Biochemical Methods

#### 4.3.1. Biochemical Blood Analysis

A detailed clinical blood test was performed on all patients, which included an assessment of 24 laboratory parameters: dynamic analysis of the indicators of the acid-base state of the blood (concentrations of calcium ions, ionized calcium (Ca^2+^), sodium ions (Na^+^), potassium (K^+^), BE (Ecf) excess bases outside the cell fluid, pH, partial pressure of carbon dioxide (pCO_2_), and bicarbonate in plasma (HCO_3_^−^)), lactate, blood oxygen saturation, and partial pressure of oxygen (pO_2_). Biochemical blood analyses included the determination of alanine aminotransferase (ALT), aspartate aminotransferase (AST), total protein, albumin, total bilirubin, direct (bound), bilirubin, glucose, lactate dehydrogenase (LDH), creatinine, and urea by an automatic hemoanalyzer XN-1000 (Sysmex Corporation, Kobe, Japan) according to the operating instructions [45]. Coagulogram parameters were also measured, e.g., APTT, fibrinogen, prothrombin time, and D-dimer. For example, APTT was evaluated using a set of reagents, STA^®^ Cephascreen^®^ (Diagnostica Stago S.A.S, Asnières-sur-Seine, France), on an STA^®^ analyzer according to the instructions [46]. The amounts of immunoglobulins (IgA, IgM, and IgG), CRP, ferritin, IL-6, and procalcitonin (PCT) were also assessed in the blood. Blood serum and plasma samples were collected in the morning on an empty stomach using vacutenirs containing Li-heparin separation gel, K2-EDTA, or K3-EDTA plasma. The blood was sent to the clinical diagnostic laboratory for examination within 1 h. The biochemical and clinical parameters and a history of comorbidities are presented in Table 8.

#### 4.3.2. Evaluation of Interleukin-6 by Electrochemiluminescent Immunoassay

The IL-6 levels in blood sera were determined using an Elecsys^®^ IL-6 electrochemiluminescent immunoassay on a cobas e602 analyzer (Roche Diagnostics Corporation, Indianapolis, IN, USA) according to the method in [47]. This test takes 18 min to measure the concentration of IL-6 in the range of 1.5–5000 pg/mL in a sample volume of 30 µL. The immunoassay is based on the sandwich principle, when a blood sample is added to immobilized microparticles coated with streptavidin. After incubation, imaging is performed using biotinylated mouse monoclonal antibodies to IL-6 (0.9 mcg/mL) diluted in a phosphate buffer of 95 mmol/L, at pH 7.3 [48].

#### 4.3.3. Evaluation of Procalcitonin Using an Immunochromatographic Test

Procalcitonin in serum and plasma was evaluated using the BRAHMS PCT-Q immunochromatographic test (BRAHMS, Hennigsdorf, Germany) according to the instructions [49]. Six drops of a blood sample were dropped into a round hole on a BRAHMS PCT-Q tablet using a pipette and incubated for 30 min at room temperature. After the specified time (maximum 45 min), the concentration range of the PCT sample was determined. At the beginning, a distinct appearance of the control band was visually recorded. Tests in which only the control band is observed were negative. In these tests, the concentration of PCT was less than 0.5 ng/mL. The tests in which the control and test strips were visualized were positive. The PCT concentration was determined by comparing the intensity of staining of the test strip with the colored stripes on the control card included in the kit. The BRAHMS PCT-Q express test obtain results with a 90–92% diagnostic sensitivity and a 92–98% specificity.

#### 4.3.4. Determination of Creatinine, C-Reactive Protein, Aspartate Aminotransferase, Lactate Dehydrogenase, and Ferritin

Creatinine, LDH, and ACT, as well as CRP and ferritin, were determined, respectively, using spectrophotometric and immunoturbidimetric methods and immunoassays of chemiluminescent microparticles (CMIA) on immunochemical analyzers (Abbott Architect cSystems™ (GMI, Ramsey, MN, USA) and Abbott AEROSET (Diamond Diagnostics Inc., Holliston, MA, USA)) according to the instructions [49,50,51,52,53,54].

#### 4.3.5. The Determination of D-Dimer Using Immunoturdodimetric Analysis

The D-dimer was quantified in venous plasma samples using the STA^®^—Liatest^®^ D-Di kit (Diagnostica Stago, Asnières-sur-Seine, France) for immunotourdodimetric analysis, according to the instructions [55]. This analysis allows one to determine the D-dimer with a sensitivity of 97.0–100% (91.6–100%), a specificity of 53.3–77% (72.9–81.9%), a negative predictive value (NPV) of 99.7% (99.2–100.0%), and a positive predictive value (PPV) of 25.5–33.8% (23.5–40.5%). The plasma samples of patients were examined in undiluted form and loaded into an analyzer, on which a test was selected to evaluate the D-dimer by selecting an option on screen. The analysis of the D-dimer in the tested plasma was automatically performed by the analyzer at a wavelength of 540 nm immediately after loading the samples. The level of D-dimer (mcg/mL) in the sample of the tested plasma was displayed on the analyzer screen. D-dimer levels are expressed in initial equivalent fibrinogen units (FEU). By definition, one FEU represents the amount of fibrinogen initially present, which leads to the observed D-dimer level. The actual amount of D-dimer is about half of the FEU.

#### 4.3.6. Assessment of the Level of Secretory Phospholipase A2 by Enzyme Immunoassay

Phospholipase A2 in blood samples was determined using the sPLA2 kit (Cayman Chemical, Ann Arbor, MI, USA) for enzyme immunoassay based on the “sandwich” method of double antibodies, according to the instructions [56]. Each well of a 96-well microplate was coated with a monoclonal antibody of human sPLA2 type IIA. Such an antibody will bind to human sPLA2 type IIA located in the well. One hundred milliliters of control and biological samples were added to each well and incubated for 2 h at room temperature on a shaker. Then, the wells of the tablet were washed four times with a washing buffer and 100 µL of secondary HRP-conjugated mouse monoclonal antibodies specific to sPLA2 were added. These antibodies were used to visualize the captured sPLA2 type IIA. The samples were incubated for 1 h at room temperature on a shaker. Excess antibodies were washed 4 times with a washing buffer. The human sPLA2 type IIA concentration was determined by measuring the enzymatic activity of HRP with the addition of 100 µL of the chromogenic substrate 3,3′,5,5′-tetramethylbenzidine (TMB). The samples were then incubated for 30 min at room temperature in the dark. The formation of a blue color was monitored. The reaction was stopped by adding 100 µL of acid, and a bright yellow product was formed, which was measured on an ELISA reader at 450 nm. The intensity of this color is directly proportional to the amount of bound HRP-streptavidin conjugate and the concentration of human sPLA2 type IIA.

### 4.4. Genetic Methods

In smears from the nasopharyngeal mucosa, the presence of RNA of the SARS-coronavirus-2 virus, as well as concomitant factors of mixed infection, i.e., RNA of influenza A and B, parainfluenza virus, respiratory syncytial virus, rhinoviruses, DNA of bokavirus and adenovirus, and metapneumovirus, was evaluated by RT-PCR. Sampling of smears was carried out with sterile swabs from both nasal entrances and the nasopharynx. Smears were transported in saline solution in sterile Eppendorf-type microprobes.

#### 4.4.1. RNA Isolation

The isolation of SARS-CoV-2 virus RNA from nasopharyngeal smears was carried out according to the instructions for the extraction reagent kit, GeneJET (Thermo Scientific, Waltham, MA, USA) RNA Purification kit, at the Magna Pure System station (Roche, Indianapolis, IN, USA) and KingFisher™ [57]. The RNA concentration was measured on a Quantus fluorimeter using a Quantifluorine RNA System (Promega, Madison, WI, USA) [58]. The RNA quality was assessed using the TapeStation 4200 system device and a Highly Sensitive RNA Video Recording Analysis kit [59]. The selection of positive samples (with Cq < 25) for further investigation was carried out using real-time PCR with one of the kits for the diagnosis of the SARS-CoV-2 virus. The excavation of samples into tablets was carried out with the use of the automated stations Xiril AG and Eppendorf epMotion 5075tc. In cases of poor RNA quality, its post-purification was carried out using the GeneJET kit (Thermofisher, Waltham, MA, USA). cDNA was made using a set of SuperScpipt, Mint-2 (Eurogene, Moscow, Russia), or a set for the synthesis of double-stranded cDNA Maxima H Minus (ThermoFisher, Waltham, MA, USA) [60,61].

#### 4.4.2. Polymerase Chain Reaction

A RT-PCR analysis was performed using reagent kits for detecting coronavirus RNA SARS-coronavirus-2 in clinical material (produced by the Pasteur Research Institute of EM), GeneFinderTM, and COVID-19 plus RealAmp (OSANG Medicine Co. Ltd., Anyang-si, Korea) on CFX96 PCR devices in a real-time detection system (Biorad, Hercules, CA, USA) [62].

### 4.5. Statistical Methods

The results are presented as the arithmetic means ± standard deviation for the sample volume n (M ± m) and in terms of the median and the first quartile. The significance level was evaluated as * *p* < 0.05; ** *p* < 0.01; *** *p* < 0.001; and **** *p* < 0.0001. Statistical data processing (descriptive statistics and graphical analysis of data relationships from different tables) was performed using the GraphPad application on the Prism 8.01 platform. The frequency characteristics of qualitative indicators (gender, degree of form and pathological processes, and complaints) were evaluated using nonparametric methods, χ^2^. A Fisher’s exact test was used to compare mortality and disease severity in groups. Differences between groups were identified using the Kruskal–Wallis test and the Mann–Whitney test as a post hoc analysis [63].

## 5. Conclusions

We have shown that the blood levels of sPLA2 and IL-6 in 158 patients increase statistically significantly in cases which eventually result in death and when patients are transferred to the ICU (as the severity of the COVID-19 infection increases), showing that IL-6 and sPLA2 can be considered as early predictors of aggravation of COVID-19 infections.

## Figures and Tables

**Figure 1 ijms-24-05540-f001:**
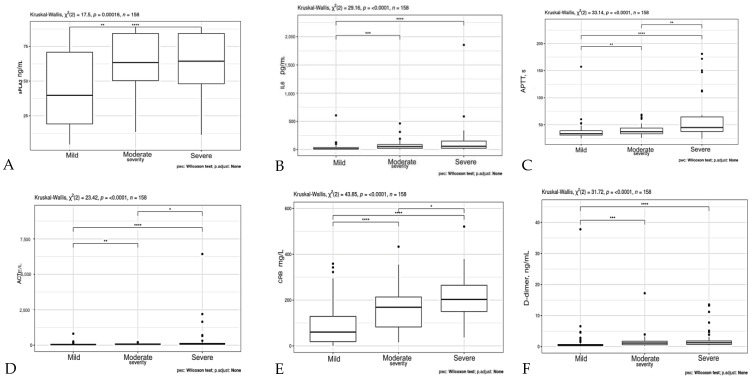
PLA2 (**A**), IL-6 (**B**), APTT (**C**), ACT (**D**), CRB (**E**), D-dimer (**F**), ferritin (**G**), LDH (**H**), procalcitonin (**I**), and GFR (**J**) levels, and hematocrit (**K**), lymphocytes (**L**), neutrophils (**M**), leucocytes (**N**), eosinophils (**O**) in patients depending on the severity of the COVID-19 infection. The results are presented as the arithmetic means ± standard deviation. The significance levels evaluated as * *p* < 0.05; ** *p* < 0.01; *** *p* < 0.001; and **** *p* < 0.0001.

**Figure 2 ijms-24-05540-f002:**
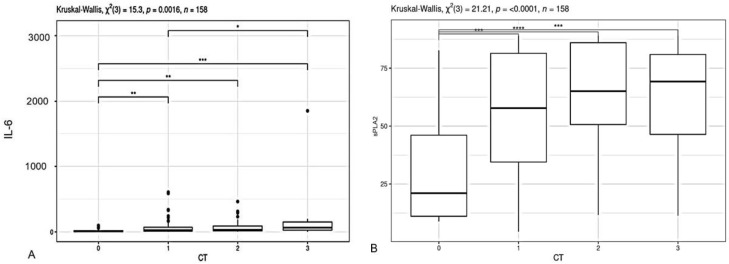
IL-6 (**A**) and PLA2 (**B**) levels in patients depending on the severity of the COVID-19 infection according to CT data. The numbers 0, 1, 2, and 3 indicate the degree of lung damage according to CT. The results are presented as medians and confidence intervals. The significance levels evaluated as * *p* < 0.05; ** *p* < 0.01; *** *p* < 0.001; and **** *p* < 0.0001.

**Figure 3 ijms-24-05540-f003:**
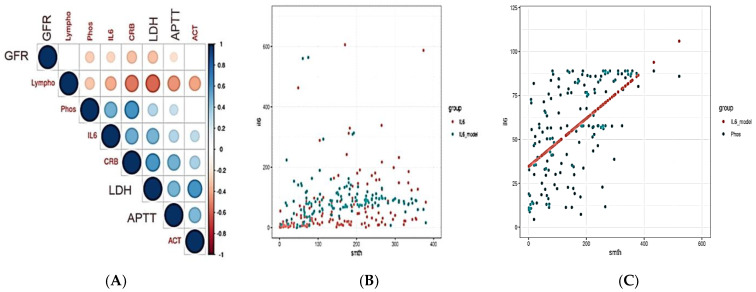
Correlation coefficients of the biochemical analytes in patients with COVID-19 (**A**) and multiple regression model for IL-6 (**B**) and PLA2 (**C**) levels.

**Figure 4 ijms-24-05540-f004:**
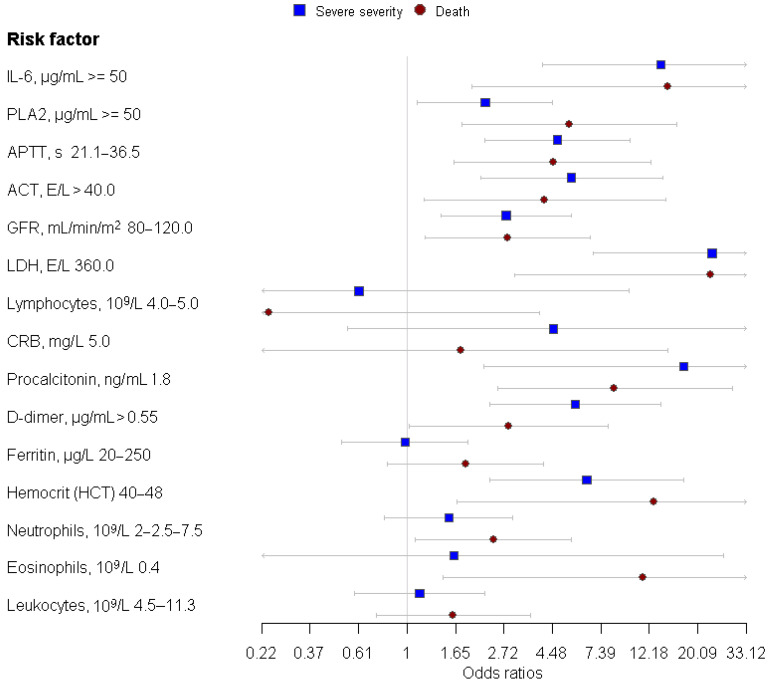
Odd ratio (OR) for patients with a severe COVID-19 infection and death.

**Figure 5 ijms-24-05540-f005:**
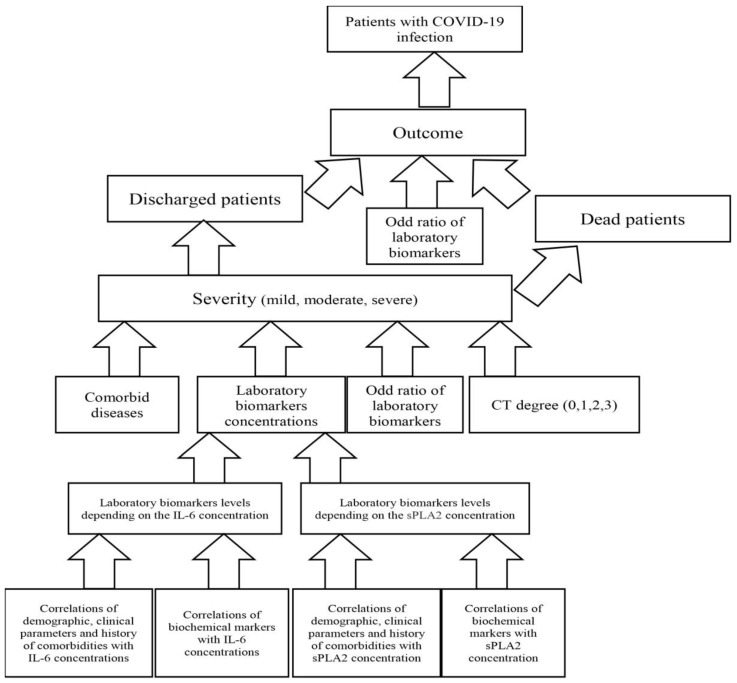
Design of the study.

**Table 1 ijms-24-05540-t001:** Distribution of patients by severity and outcome of COVID-19 infection.

Group	Patients	Men	Women	Age, Years	Men Age, Years	Women Age, Years
Mild	61	37	24	50.1 ± 12.6	52.0 ± 10.3	47.0 ± 15.3
Moderate	37	24	13	52.1 ± 11.0	51.2 ± 10.1	53.7 ± 12.7
Severe	60	37	23	51.3 ± 11.3	51.0 ± 10.4	51.9 ± 12.7
Outcome						
Survival	127	80	47	50.1 ± 11.8	50.7 ± 10.1	49.1 ± 14.3
Death	31	18	13	54.6 ± 10.6	54.5 ± 10.3	54.8 ± 11.4

**Table 2 ijms-24-05540-t002:** Changes in clinical parameters depending on the severity of COVID-19 infection.

Significative	Severity			Significance, *p*		
	Mild	Moderate	Severe	Mild vs. Moderate	Mild vs. Severe	Moderate vs. Severe
ICU-COVID-19, shares units	**0.31 ± 0.06**	**0.81 ± 0.06**	**1.0 ± 0.0**	**0.0001**	**0.0001**	**0.0102**
Days from illness to biobanking, days	**6.8 ± 0.56**	**9.9 ± 0.71**	**12.1 ± 0.78**	**0.0097**	**0.0001**	**0.0434**
CT max	**0.76 ± 0.05**	**2.0 ± 0.00**	**3.4 ± 0.06**	**0.0001**	**0.0001**	**0.0001**
Chronic heart failure	0.31 ± 0.06	0.24 ± 0.07	0.30 ± 0.06	0.8290	0.8425	0.8290
Coronary artery disease	0.25 ± 0.05	0.22 ± 0.07	0.25 ± 0.05	0.9754	0.9999	0.9754
Cardiac arrhythmias	0.12 ± 0.04	0.027 ± 0.027	0.03 ± 0.023	0.1752	0.1752	0.9013
Valvular disease	0.017± 0.016	0.027 ± 0.027	0.017± 0.016	0.9781	0.9781	0.9999
Pulmonary circulation disorders	**0.067 ± 0.032**	**0.24 ± 0.07**	**0.47 ± 0.06**	**0.0395**	**0.0001**	**0.0188**
Pressure in the pulmonary artery	0.08 ± 0.04	**0.30± 0.09**	**0.73 ± 0.11**	0.1145	**0.0001**	**0.0030**
Peripheral vascular disorders	**0.0 ± 0.00**	0.297 ± 0.09	**0.033 ± 0.02**	**0.0001**	0.5408	**0.0001**
Arterial hypertension, uncomplicated	**0.0 ± 0.00**	0.27 ± 0.07	**0.37 ± 0.06**	**0.0013**	0.2161	**0.0001**
Arterial hypertension, complicated	0.47 ± 0.06	0.43 ± 0.08	0.62 ± 0.06	0.7427	0.2181	0.2181
Coagulopathy	0.10 ± 0.039	0.19 ± 0.06	**0.33 ± 0.06**	0.2855	**0.0048**	0.1632
Iron deficiency anemia	0.36 ± 0.06	0.51 ± 0.08	**0.68 ± 0.06**	0.1822	**0.0014**	0.1822
Varicose disease	0.07 ± 0.03	0.11 ± 0.05	0.15 ± 0.04	0.7706	0.3749	0.7706

Note: Statistically significant (*p* < 0.05) differences in the groups are highlighted in bold. The results are presented as arithmetic means ± standard deviations. Numbers, such as 0.31 for chronic heart failure, represent the ratio of the number of patients with this disease to the total number of patients with a such degree of severity. For example, 0.31 means that chronic heart failure was observed in 19 patients with mild severity (0.31 × 61), where 61 is the number of patients with mild severity.

**Table 3 ijms-24-05540-t003:** Laboratory analyte levels depending on the IL-6 and PLA2 concentration.

Analyte		Significance, *p*	
		from IL-6	from sPLA2
APTT, s	**50.50 ± 37.47**	**<0.0001**	0.1634
ACT, E/L	154.4 ± 43.79	0.1865	**0.0253**
GFR, mL/min/m^2^	**75.05 ± 28.39**	**<0.0001**	**<0.0001**
LDH, E/L	**609.5 ± 65.44**	**<0.0001**	**<0.0001**
Lymphocytes, 10^9^/L	**0.9351 ± 0.5341**	**<0.0001**	**<0.0001**
CRB, mg/L	153.6 ± 108.7	0.0534	**<0.0001**
Procalcitonin, ng/mL	**1.037 ± 3.234**	**<0.0001**	**<0.0001**
Hematocrit, %	**33.13 ± 12.37**	**<0.0001**	**<0.0001**
D-dimer, ng/mL	**1.805 ± 3.823**	**<0.0001**	**<0.0001**
Ferritin, mkg/L	**725.8 ± 104.2**	**<0.0001**	**<0.0001**
Neutrophils, 10^9^/L	**5.020 ± 3.405**	**<0.0001**	**<0.0001**
Leukocytes, 10^9^/L	**7.464 ± 5.122**	**<0.0001**	**<0.0001**
Eosinophils, 10^9^/L	**0.04120 ± 0.06989**	**<0.0001**	**<0.0001**

Note: bold values indicate statistically significant (*p* < 0.05) differences in groups. The results are presented as the arithmetic mean ± standard deviation.

**Table 4 ijms-24-05540-t004:** Coefficients of correlations of IL-6, sPLA2, demographics, clinical parameters, and history of comorbidities.

Significative	Correlation Coefficient, r		Significance, *p*	
	IL-6	sPLA2	IL-6	sPLA2
ICU-COVID-19	**0.502622**	**0.259144**	**1.7 × 10^−11^**	**0.001009**
Gender (one male and two female)	−0.10282	0.071739	0.198587	0.370395
Age group (one under 45, two 45–60, and three over 60)	0.049938	0.103635	0.53321	0.195037
Height, cm	0.135512	−0.04525	0.08957	0.572402
Weight, kg	**0.160815**	0.095017	**0.043537**	0.235011
CT maximum	**0.618787**	**0.378456**	**0.0**	**9.44 × 10^−7^**
Chronic heart failure	0.019873	−0.00695	0.804261	0.930937
Coronary artery disease	0.068477	0.068667	0.392602	0.391287
Cardiac arrhythmias	−0.03952	−0.08416	0.621986	0.293092
Valvular disease	0.087955	0.063152	0.271797	0.430526
Pulmonary circulation disorders	**0.398092**	0.11856	**2.22 × 10^−7^**	0.137894
Pressure in the pulmonary artery	**0.399486**	0.087964	**2 × 10^−7^**	0.27175
Peripheral vascular disorders	**0.165059**	−0.00699	**0.038217**	0.93052
Arterial hypertension, uncomplicated	0.11885	0.074727	0.136927	0.350742
Arterial hypertension, complicated	0.099637	0.104632	0.212923	0.190754
Paralysis	−0.06702	−0.094	0.402799	0.240082
Other neurological disorders	0.078687	0.12773	0.32573	0.109742
Chronic lung disease	−0.01431	−0.00949	0.858333	0.905834
Diabetes mellitus, uncomplicated	**0.245662**	**0.205014**	**0.001863**	**0.009765**
Diabetes mellitus, complicated	0.035367	0.089572	0.659092	0.263047
Hypothyroidism	−0.07941	0.044169	0.321266	0.581595
Kidney failure	**0.187362**	0.100827	**0.018407**	0.207481
Liver disease	0.040845	−0.09583	0.610363	0.231022
Peptic ulcer without bleeding	0.002595	0.084268	0.974183	0.292481
Solid tumor without metastases	−0.13941	0.009944	0.080631	0.901312
Rheumatoid arthritis/collagen/vascular diseases	**0.199602**	0.049235	**0.011924**	0.538997
Coagulopathy	**0.32484**	0.036364	**3.12 × 10^−5^**	0.650113
Obesity	0.135221	**0.181496**	0.090267	**0.022475**
Weight loss	−0.08131	−0.06064	0.309786	0.449095
Fluid and electrolyte disorders	−0.0425	−0.05379	0.595994	0.502099
Iron deficiency anemia	**0.322062**	**0.320441**	**3.68 × 10^−5^**	**4.05 × 10^−5^**
Alcohol abuse	−0.11536	0.005502	0.148906	0.945304
Varicose disease	0.126784	0.089505	0.112421	0.263408

Note: bold values indicate statistically significant (*p* < 0.05) differences between groups. Correlation coefficients for binary indicators, such as coronary heart disease, were calculated using the ratio of the number of patients with this disease to the total number of patients.

**Table 5 ijms-24-05540-t005:** PLA2 and IL-6 correlation coefficients with biochemical parameters.

Analyte	Correlation Coefficient, r		Significance, *p*	
	PLA2	IL-6	PLA2	IL-6
IL-6 levels on the date of BB (median 30.24)	**0.382558**	**0.391871**	**7.03 × 10^−7^**	**3.55 × 10^−7^**
IL-6 levels after the date of BB (1: under 40 and 2: over 40 pg/mL)	**0.408909**	**0.457899**	**9.6 × 10^−8^**	**1.46 × 10^−9^**
IL-6 maximum per hospitalization	0.15323	**0.437241**	0.054588	**9.24 × 10^−9^**
IL-6 max group *	**0.42782**	-	**2.06 × 10^−8^**	**0**
IL-6 max-group (1: up to 40 and 2: more than 40 pg/mL)	**0.350996**	**0.543498**	**6.14 × 10^−6^**	**1.6 × 10^−13^**
IL-6 **	**0.447366**	**0.581317**	**3.79 × 10^−9^**	**1.11 × 10^−15^**
PLA2, ng/mL	-	**0.427819**		**2.0606 × 10^−8^**
APTT, s	0.15005	**0.336515**	0.059865	**1.54 × 10^−5^**
ACT, E/L	0.034423	**0.220623**	0.667648	**0.005343**
GFR, mL/min/m^2^	**−0.22393**	**−0.30408**	**0.004678**	**0.000103**
LDH, E/L	0.148297	**0.394712**	0.062948	**2.87 × 10^−7^**
Lymphocytes, 10^9^/L	**−0.30304**	**−0.54144**	**0.000108718**	**2.05 × 10^−13^**
CRB (quantitative), mg/L	**0.562617**	**0.597305**	**1.42109 × 10^−14^**	**0**
Laboratory score at admission	**0.528211**	**0.773774677**	**9.85 × 10^−13^**	**0**

Note: Statistically significant correlations at *p* < 0.05 are highlighted in bold. * IL-6 group (1: up to 100 pg/mL; 2: 100–200 pg/mL; 3: more than 200 pg/mL); ** group IL-6 (1: up to 40 and remained up to 40 pg/mL; 2: up to 40 and rose to more than 40 pg/mL; 3: more than 40 and remained more than 40 pg/mL).

**Table 6 ijms-24-05540-t006:** The outcome of the disease depending on the severity of the COVID-19 infection.

Significative	Severity			Significance, *p*		
	Mild	Moderate	Severe	Mild vs. Moderate	Mild vs. Severe	Moderate vs. Severe
Death (0: recovery, 1: death), fractions of units	**0.016 ± 0.016**	**0.054 ± 0.03**	**0.46 ± 0.06**	0.5997	**0.0001**	**0.0001**

Note: bold values indicate statistically significant (*p* < 0.05) differences in groups. The results are presented as the arithmetic means ± standard deviation. Numbers, such as 0.016, represent the ratio of the number of dead patients to the total number of dead patients at each severity degree.

**Table 7 ijms-24-05540-t007:** Analysis of laboratory analytes depending on the outcome of the disease.

Analyte	Survival	Death	Kruskal–Wallis	Significance, *p*
IL-6, pg/mL	25 (10–66)	127 (29–183)	**0.0**	**0.00042**
PLA2, pg/mL	55 (30–76)	81 (58–86)	**0.0001**	**0.00079**
APTT, s	**37 (32–44)**	**48 (38–89)**	**0.0001**	**0.0**
ACT, E/L	**59 (38–96)**	**109 (59–142)**	**0.0012**	**0.001**
LDH, E/L	**389 (282–537)**	**726 (622–1199)**	**0.0000**	**0.0**
GFR, mL/min/m^2^	**84 (65–96)**	**51 (18–85)**	**0.0001**	**0.0**
Lymphocytes, 10^9^/L	**1 (1–1)**	**0 (0–1)**	**0.0000**	**0.0**
CRB, mg/L	**110 (57–202)**	**259 (186–331)**	**0.0000**	**0.0**
PCT, ng/mL	0 (0–1)	1 (0–1)	0.0853	0.086
Hematocrit, %	**38 (33–42)**	**33 (19–37)**	**0.0004**	**0.0**
D-dimer, ng/mL	**1 (0–2)**	**1 (1–2)**	**0.0029**	**0.003**
Ferritin, mkg/L	**268 (115–850)**	**1156 (160–1782)**	**0.0088**	**0.009**
Neutrophils, 10^9^/L	**4 (3–6)**	**6 (4–9)**	**0.0033**	**0.003**
Eosinophils, 10^9^/L	0 (0–0)	0 (0–0)	0.1950	0.196
Leukocytes, 10^9^/L	6 (5–8)	7 (5–12)	0.0818	0.082

Note: bold values indicate statistically significant (*p* < 0.05) differences in groups. The results are presented as medians and confidence intervals.

**Table 8 ijms-24-05540-t008:** Biochemical and clinical parameters and history of comorbidities.

Analyte	Clinical Parameters and History of Comorbidities
IL-6, pg/mL	Outcome
PLA2, pg/mL	Degree of severity
APTT, s	Degree on CT data
ACT, E/L	ICU-COVID-19, shares units
LDH, E/L	Days from illness to biobanking
GFR, mL/min/m^2^	Gender
Lymphocytes, 10^9^/L	Age
CRB, mg/L	CT max
PCT, ng/mL	Chronic heart failure
Hematocrit, %	Coronary artery disease
D-dimer, ng/mL	Cardiac arrhythmias
Ferritin, mkg/L	Valvular disease
Neutrophils, 10^9^/L	Pulmonary circulation disorders
Eosinophils, 10^9^/L	Pressure in the pulmonary artery
Leukocytes, 10^9^/L	Peripheral vascular disorders
Laboratory score	Arterial hypertension, uncomplicated
	Arterial hypertension, complicated
	Coagulopathy
	Iron deficiency anemia
	Varicose disease

## Data Availability

Not applicable.

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
