# Peer review of "Secretory Phospholipase A2 and Interleukin-6 Levels as Predictive Markers of the Severity and Outcome of Patients with COVID-19 Infections"

_ijms, 2023, doi:10.3390/ijms24065540_

Round 1

Reviewer 1 Report

Well written manuscript, i must congratulate all the authors. However few modification are needed for better understanding. Comments are given in manuscript as sticky notes.

Author Response

Dear reviewer,

Thank you very much for your comments and suggestions for improving our article. We took into account your comments and made corrections to the text of the article, highlighting it in color.

Reviewer 2 Report

Your title and abstract interested me considerably as our lab too has invested in searching for biomarkers that will help with diagnosis and treatment recommendations for COVID-19.

Your introduction is very well stated and clear (although I will make some detailed comments on it below). 

However, the results section needs some work, particularly in the reporting of the results in the Tables, which I found confusing overall. 

My intuition on the question aligns with your conclusions. Higher IL-6 and sPLA-2 should correlate with more severe disease.

IL-6 as a Specific COVID-19 Biomarker

We all are aware that IL-6 is a general inflammation marker. For all diseases. The more inflamed the body or a specific site, the higher will be the IL-6 level. You should include something about how IL-6 specifically stands out for COVID. It could be, for example, that a COVID patient with another condition (like being HIV or HCV positive) will have elevated IL-6 levels due to that condition and not the COVID. How do you distinguish between the causes of IL-6 elevation. 

PLA-2 as a Biomarker for COVID

On lines 90 - 97, you indicate that this enzyme is a marker related to a number of diseases. Same question. What is the specific relationship to COVID?

Ct vs CT

In many places you refer to "CT" as a measure. With regard to COVID, I normally think of cycle threshold for the PCR test that indicates viral load (higher number, lower viral load, etc.). But you also speak of computerized tomography (CT). Which meaning do you intend? And, if it's tomography, what is actually being measured? and what does change in this represent?

The Real Table 1

Your Table 7 where you give the demographic characteristics of your participants really should be Table 1. As a reader and evaluator, I want to know at the start who is being studied and are they an adequate sample that represents a population I find of interest. Please insert that Table before your Table 1.

Table 1 and Table 2

As a methodologist, I see many problems with these tables. First, you use indices of change in the various conditions you measure rather than real values. I found this confusing. What does Chronic Heart failure of 0.32 actually mean. What is it the difference of? Likewise on the other categories. Also, I'm not sure BMI, either in absolute numbers or this degree of obesity scale, means much. I think you can drop these measures with no implications. I have not seen any COVID studies where this is significant. Your indication  in these tables of margin of error +/- relates to what margin? to simple deviation or to a normal distribution based measure like standard error or standard deviation?

Figure 1 (and those like it)

You will note that the panels don't quite fit in the spaces you have reserved for them.

Table 3

Here you use a more standard presentation of a mean with what appears to be a range. Across your tables, the measure of deviation should be standard. Decide on 1 and stick to it. However, what do the categories of CT data mean? 0 - 3 what units?

Tables 4 and 5

Here you categorize levels of IL-6 into 3 categories. Why. IL-6 is a good numeric variable and categorization results in the loss of information. Why not simply measure APTT for example against IL-6 levels numerically and do a t-test of the difference in means or a Wilcoxon test in the difference in medians? Same thing for levels of PLA2 in Table 5.

Table 7 (separate issue)

Here you categorize numeric variables like age group. Why? Age is a good numeric variable allowing all sorts of statistical tests. You also have correlations between variables I don't see as being numeric (like Coronary artery disease, which is surely binary -- you have it or you don't). You need to explain what you are calculating as a correlation coefficient there. Same thing with all the other comorbidities you list in that table.

Figure 7 -- Odds Ratio

You start your measure of odds ratio at 1 and go up from there? There were none that were below 1? Doesn't make a lot of intuitive sense. Also the format of your graph is awkward. Try to use a Forest Plot here.

Section 4.5 Statistical Methods

You bury here right at the end information that should be in the caption of your graphs and tables related to means +/- standard deviation.

My Conclusion

You are covering important territory, but need to do some additional work to make this an outstanding article. Also, you need more focus on the statistics and clarify their presentation. p-values are not as important as a clear description of what quantities you are measuring. Not unimportant, but I want to know what you are measuring. Finally, have a native English speaker go over the text to clean up some of the words that appear to me (a native speaker) as literal translations from Russian.

Author Response

Dear reviewer,

Thank you very much for your comments and suggestions for improving our article.

Reviewer 3 Report

Urazov et al. contributed in the manuscript entitled " SECRETORY PHOSPHOLIPASE A2 AND INTERLEUKIN-6 LEVELS AS PREDICTIVE MARKERS OF THE SEVERITY AND OUTCOME OF PATIENTS WITH COVID-19 INFECTION" which is informative and will have significant impact in the field of Molecular Pathology, Diagnostics and Therapeutics for COVID-19. 

The authors' results are original, and they aid the scientific community in early COVID-19 diagnosis using PLA2 and IL-6 as biochemical indicators.

Author Response

Dear Reviewer,

Thank you very much for your opinion and review of our article.

Round 2

Reviewer 1 Report

please provide final revised manuscript ( no highlightings in manuscript)